## Registered report

psychology/cognition

mind perception, the Medusa effect, reality, prosocial behaviour, dictator game

**Author for correspondence:**
Jing Han
e-mail: janehan1995@gmail.com

# The Medusa effect: a registered replication report of Will, Merritt, Jenkins and Kingstone (2021)

Jing Han[1], Minjun Zhang[1], Jiaxin Liu[1], Yu Song[1] and Yuki Yamada[2]

[1]Graduate School of Human-Environment Studies, and [2]Faculty of Arts and Science, Kyushu University, Fukuoka, Fukuoka, Japan

 JH, 0009-0001-9270-1677; MZ, 0000-0002-5431-1686;
JL, 0009-0005-5270-4629; YS, 0009-0008-1000-1636; YY, 0000-0003-1431-568X

Will *et al.*'s (2021 *Proc. Natl Acad. Sci. USA* 118, e2106640118 (doi:10.1073/pnas.2106640118)) found the Medusa effect, which refers to the tendency that people evaluate a 'person in picture' more mindful than a 'person in picture of a picture'. The present study tried to directly replicate the Experiments 2 and 5 of Will *et al.*'s (2021), to examine the reliability, validity and generalization of the Medusa effect, as well as its effect on prosocial behaviour. We used the same stimuli and methodology as the original research, but recruited participants in Japan with a larger sample size ($N = 1387$ in total) as a registered report. In our two replication experiments, we again found that pictures with lower levels of abstraction (L1) were perceived to convey more mind and lead to higher levels of prosocial behaviour, successfully replicating the original findings. The results of the present study suggested the high reproducibility and generalizability of the Medusa effect. Pre-registered Stage 1 protocol: https://osf.io/xj46z (date of in-principle acceptance: 9 February 2023)

## 1. Introduction

The capacity to bridge the divide between one's own mind and others' minds is one of the most useful tools in social life. People use this impressive tool to understand, predict, and even control others' behaviour and develop social connections with them [1]. During this process, most researchers have focused on how people perceive others' mental states, which is often referred to as 'mind perception'.

As pictures play an essential role in not only containing features but also expressing emotions, mind perception involving pictures is

a theme worthy of research. Focusing on the aspect of realism, a person's image in a picture would seem less realistic than their actual presence, as it often lacks certain aspects that are lost in the process of pictorial abstraction. For example, a real person is more mindful than a portrait in a gallery. Will *et al.* [2] found the tendency of people to evaluate a 'person in picture' as more mindful than a 'person in picture of a picture', and named it 'the Medusa effect'. This phenomenon is intriguing as it suggests that when evaluating the humanness of others, people may integrate and interpret information based on the context or level of abstraction in which it is presented. In other words, the context in which a person is perceived— whether in real life or within a photograph—significantly influences how we perceive and interpret their humanness. The abstraction of information, or how it is presented and perceived in different contexts, plays a crucial role in this process.

## 1.1. The Medusa effect

Will *et al.* [2] describe the 'Medusa effect' as a phenomenon where mind perception diminishes with increased abstraction, drawing on the Greek myth of Medusa. In this myth, Medusa, a figure whose gaze turned those who directly saw her into stone, was defeated by Perseus. Perseus's use of a reflective shield to view Medusa indirectly can be metaphorically related to the concept of abstraction in mind perception. While Medusa herself represents a direct, less abstract encounter (L1), the reflection in Perseus's shield, akin to viewing a picture, symbolizes a more abstract level (L2) where mind perception is reduced. Besides this mythology story, researchers have found that pictures can convey more than they imagined. For example, pictures with eyes can reflexively shift our attention elsewhere [3]. Furthermore, it has been reported (although reproducibility is debatable: e.g. [4,5]) that being gazed at with eyes in pictures increases prosocial behaviour [6]. For instance, pictures of eyes can increase generosity in dictatorial games [7], charitable giving in the field [8], as well as cooperative behaviour in the field setting [9]. The mind perception thus conveyed, leads people to help or hurt others or to praise and punish others in their lives. Precisely because mind perception is the ability to reason about thoughts, numerous researchers have focused on how mind perception is defined, when it occurs and its importance.

A considerable amount of research has proposed various frameworks for quantifying mind perception and its dimensions [10,11]. However, the most influential one is the two-dimensional framework proposed by Gray *et al.* [12], which refers to people intuitively thinking about other minds in terms of Experience (the capacity to sense and feel) and Agency (the capacity to plan and act). The original research on the Medusa effect adopted this framework and explored the differences in mind perception among different levels of pictorial abstraction.

Although mind perception occurs in the perceiver's mind, the characteristics of the entity being perceived also influence mind perception [1]. Therefore, the original study focused on the effects of different abstractions on mind perception of pictures. Pictures, as important bearers of expressions of people's ideas and perceptions of their authenticity, are uploaded billions of times daily on the Internet [13]. Moreover, an important feature of pictures caught the attention of the original researchers: they may not only contain partial information about reality, but may also contain pictures that serve as different levels of abstraction [2]. In addition, the higher the abstraction level of a picture, the lower the potency of the subject. According to the above speculation, the original study primarily hypothesized that different levels of abstraction would bring about different levels of mind perception.

Following the aforementioned previous research, Will *et al.* [2] used five experiments to verify the existence of the Medusa effect. First of all, it is necessary to clarify that the original study specified the real person as L0 ('L' is short for level, referring to the levels of abstraction), the picture of a person as L1, and the picture containing a picture of a person as L2, referring to different pictorial abstraction levels. Based on this setting, the original Experiments 1 and 2 tested the central hypothesis of the original study that L1 brings more mind perception than L2, by means of choice and evaluation. The original Experiment 3 conducted a free viewing task by using the same stimuli as Experiments 1 and 2 and monitored the looking behaviour of participants via an eye tracker. The eye-tracking data showed that differentiation between L1 and L2 occurs spontaneously during passive viewing, even without an explicit mind perception task. Experiment 4 improved the stimuli to replicate the picture abstraction cost and further compared transitions between different levels of representation (L0 to L1, and L1 to L2). Finally, Experiment 5 used the dictator game to verify the possible effects of the Medusa effect on prosocial behaviour by using the newly designed stimuli in Experiment 4. In previous research, dictator games were frequently used in various studies investigating social norms, such as altruism and fairness [14], or as an experimental method to measure prosociality. The main

process of the game is that the participant controls a sum of money and decides how much money to give to the recipient; the remaining money goes to the participant. Furthermore, many studies have demonstrated that the amount of money distribution in the dictator game is significantly correlated with the social salience of the recipient [14,15]. Experiment 5 linked behaviour and cognition through the dictator game, verifying that picture abstraction would act on the game process through mind perception leading to differences in allocation amounts.

In conclusion, by conducting these five experiments, Will *et al.* [2] first demonstrated the hypothesis that the abstraction level increases from L0 to L1 and from L1 to L2, thereby decreasing the level of mind perception associated with pictures. In addition, for the first time, the 'Medusa effect' in mind perception was demonstrated, suggesting a new focus on psychological effects in a modern environment, where many emotions are communicated through the Internet and pictures. Moreover, mind perception plays a pivotal role in major areas of society such as education, law, medicine and charity [16]. Inevitably, it is necessary to consider the different behaviours resulting from varying degrees of mind perception through pictorial abstraction, which has attracted our attention.

## 1.2. Aims of the present study

Will *et al.* [2] conducted five experiments to determine the degree of mind perception between different levels of pictorial abstraction and its prosocial effects. For the first two experiments, we decided to replicate Experiment 2 rather than Experiment 1, to confirm the existence of the effect of using a rating task, which is more informative than the two-alternative forced choice (2AFC) task in Experiment 1 of the original research. As for Experiment 3, eye-tracking data showed differentiation between L1 and L2 occurs spontaneously, and it explored the mechanism of the effect. In reference to Experiment 4, Experiment 3 replicated the effect and further compared transitions by adding a new condition of a real person (L0). Since both Experiments 3 and 4 were conducted in the laboratory and the COVID-19 pandemic is still in process, it is hard for us to replicate them at this time. Furthermore, the Medusa effect itself refers to how the different abstraction levels of pictures affect mind perception. Consequently, we decided to not conduct Experiment 4 but only focus on the stimuli of pictures in our replication. Additionally, we will replicate Experiment 5 to explore whether different pictorial abstraction levels influence behaviour in social interactions. Experiment 5 used the same pictures as Experiment 4 with better control of irrelevant variables (e.g. equated facial appearance across different abstraction levels and matched same image size) to conduct a mind perception task and the dictator game [2].

The original research was the first to reveal the Medusa effect and explore its prosocial effects, showing that the abstraction levels of the picture itself can reduce mind perception. This finding of the abstraction cost between different levels of pictorial abstraction suggests a novel hypothesis for future researchers, namely that people's cognitive effects may be weakened by a higher level of pictorial abstraction; in this research, for example, mind perception is reduced. We noted that, since pictorial representation has been playing a vital role in providing information on the Internet [13], the differences in abstraction levels could affect considerable social interactions by influencing mind perception; for instance, they may affect online experiments involving pictorial stimuli of faces.

Furthermore, whether this effect can be replicated in a different group of participants is also important for its generalization. The original dictator game data suggest that the susceptibility to the Medusa effect may vary among different people [2]. This result may reflect individual differences in underlying cognitive abilities, such as face perception ability [17]. Moreover, when it comes to the cross-culture effect on mind perception, according to Krumhuber *et al.* [18], when participants evaluate mind perception of faces that range on a continuum from real to artificial, intergroup processes (i.e. in-group favouritism and out-group dehumanization) play a key role in humans' perception. To be specific, for instance, participants from India evaluated South Asian (in-group) faces more mindful than Caucasian (out-group) faces [18]. This suggests that a similar influence may also exist when it comes to pictorial abstraction perception. Since the original stimuli are 'out-group' faces for participants in Japan, based on the previous research, the cross-culture effect may be one of the reasons to account for less sensitivity of mind perception.

For the above reasons, in this research, we aim to replicate Experiments 2 and 5 of the original research, in Japan, to examine the existence and generalization of the Medusa effect and its prosocial effects. Moreover, in terms of sample size, 564 participants will be recruited for Study 1 and 660 will be recruited for Study 2, based on a prior power analysis.

Based on the above review, we plan to test the following hypotheses. For H1 (in Study 1), participants would rate L1 as having higher levels of both Realness, Agency and Experience than L2. Study 2 will present three hypotheses, H2-a, H2-b and H2-c, which refer to the 2AFC task, the dictator game, and the final individual difference analysis. Regarding H2-a, participants would choose L1 rather than L2 in all three Realness, Agency and Experience dimensions. As for H2-b, in the dictator game, participants would allocate more money in condition L1 than in condition L2. With respect to H2-c, the above analysis of individual differences in perception and behaviour allows for the attribution of effects in the dictator game to the mind perception task. Accepting all three sub-hypotheses of Study 2 will be confirmed as support for H2 and the successful replication of Experiment 5.

If both H1 and H2 are supported, it confirms the original claim, with its reproducibility and generalizability extended. Picture abstraction cost and its behaviour effect exist across different races of stimuli and participants. If H1 is supported, but H2 is not, the Medusa effect may exist but is not related to prosocial behaviours. We will explore the possible reason by conducting Study 3-b with the equated race between stimuli and participants. If H2 is supported, but H1 is not, the result may be caused by the limitations of the stimuli, since the stimuli of Study 2 are newly made presentations with better control of irrelevant variables, including equated facial appearance and the same image size. By contrast, the stimuli of Study 1 are pictures downloaded from the Internet that vary in terms of size, quality, gender, race and emotional expression. Alternatively, the Medusa effect may have stronger consequences for implicit behaviour. If neither H1 nor H2 are supported, the reproducibility of the Medusa effect fails. We will redesign the stimuli with the same ethnicity as the participants and further attempt additional replications to explore possible reasons for the failure.

We plan to conduct the conditional study (Study 3) only if we have not replicated Studies 1 or 2 successfully, by using newly designed stimuli with higher quality and the same ethnicity of participants to further replicate the original Experiments 2 or 5. In the case that H1 is not supported, we will conduct Study 3-a to test H1 again. If Study 2 fails to replicate the original Experiment 5, we will conduct Study 3-b, hoping to conduct a further replication with newly constructed stimuli. If both Studies 1 and 2 fail to replicate the original experiments, we will use improved stimuli to conduct additional replication, referring to Study 3 (a and b), further examining the Medusa effect.

# 2. Study 1

Study 1 employed the mind perception assessment task to examine whether pictorial abstraction levels differ in the degree of mind perception, which was a direct replication of Will *et al.*'s [2] Experiment 2, to examine the existence of the Medusa effect in Japan.

## 2.1. Method

### 2.1.1. Independent variable

#### 2.1.1.1. Different levels of abstraction of people's photos

There were two abstraction levels in our study: L1 (picture of a person) and L2 (picture of a picture of a person). Similar to Will *et al.*'s [2] study, pictorial abstraction was a within-subjects factor.

### 2.1.2. Dependent variable

#### 2.1.2.1. Mind perception

In Study 1, mind perception was measured based on three dimensions, using the quantitative framework of mind perception that Will *et al.* [2] used from 0 (lowest level) to 10 (highest level). Participants were randomly assigned into three groups, referring to the different tasks of assessing the Realness, Agency and Experience of persons in L1 and L2.

### 2.1.3. Participants

#### 2.1.3.1. Sample size and power analysis

At least 564 participants needed to be recruited for Study 1 based on a prior power analysis using G*Power 3.1.9.7 for Windows [19,20] to replicate the Medusa effect of the study by Will *et al.* [2]. In

their study, 320 participants were randomly assigned into three groups to rate Realness ($n = 107$), Experience ($n = 109$) and Agency ($n = 104$) of the pictures. Three paired $t$-tests were conducted independently for each rating group to compare L1 and L2. The results of the analysis revealed that all the groups, Realness, Experience and Agency, of L1 compared with L2 reached significant differences (all $ps < 0.05$), and for each of the effect size, Cohen's $d$ was given ($d_R = 0.83$, $d_E = 0.34$, $d_A = 0.39$). However, the findings and statistical results in their study were the latest research developments, and other reference studies were rarely available. In addition, the effect size could sometimes be overestimated owing to the small sample size and tended to decrease in subsequent replication studies, which was a statistical bias named 'Winner's Curse' [21–23]. Based on these two points, we planned to use a small effect size ($d = 0.2$), which was defined by Cohen [24]. We conducted a one-tailed, paired $t$-test power analysis by assuming Cohen's $d = 0.2$ as small effect size, significance level $\alpha = 0.05$, and power level $1 - \beta = 0.80$ [25] to calculate our sample size. The results indicated that 156 participants will be required per group (i.e. 468 in total). In addition, it was conceivable that a certain number of participants may withdraw from the experiment midway due to dissatisfaction; hence, we added approximately 20% to this number in case of power loss due to data exclusions (i.e. 562 in total). To equalize the number of recruitments for the three groups, we added two to this number (i.e. 564 in total).

### 2.1.3.2. Recruitment and screening

We recruited participants in Japan via the Yahoo! Crowdsourcing Service (https://crowdsourcing.yahoo.co.jp/). All participants completed the study online in exchange for monetary compensation. All questions needed to be filled out for the participants to submit the answer, and we screened the participants' IP addresses to preclude repeat submissions. Data continued to be collected until a minimum sample size of 564 was reached, as indicated by the sample size analysis. Considering that it was difficult for us to limit the number of participants to exactly 564 due to the characteristics of the participatory online recruitment system, we recruited at least 564 participants and used their data for the analysis based on the timestamp.

### 2.1.3.3. Stimuli and design

The original stimuli (pictures) were provided to us via email by the original authors on request. We used the same stimuli as in the original research. The stimuli contained a total of 29 pictures, and each picture depicted an L1 person and an L2 person, which presented different degrees of abstraction in a single scene. For example, a person (L1) holding a portrait photograph (L2), or a computer user (L1) with an onscreen interlocutor (L2). The depicted L1 and L2 of each scene varied in terms of size, quality and on-screen location (left or right), and the depicted L1 and L2 persons also varied in age, gender, race and emotional expression. All the scenes were cropped to a standard size of 400 pixels high × 600 pixels wide.

Pictorial abstraction (L1 and L2) was a within-subjects factor. Participants were randomly assigned to one of the Experience, Agency or Realness conditions, which referred to their evaluation task of both L1 and L2, to examine whether there was a difference in mind perception between the different pictorial abstractions.

### 2.1.4. Procedure

The participants read the instruction (Japanese-translated version of the original one) and provided informed consent before participating in the study. They were informed that they could withdraw participation at any time. We did not collect any personal information except for gender and age. The collected data were strictly protected.

Our study strictly followed the same procedures used in the study by Will et al. [2], except for using the Yahoo! Crowdsourcing Service to recruit participants in Japan and presenting the instructions and questions in Japanese.

At the beginning of the experiment, demographic information on the participants' age and gender were collected. Thereafter, the participants were shown pictures consisting of two people with different abstraction levels (L1 and L2). Their task was to rate each of the two people based on Experience, Agency or Realness. For each trial, a single picture was shown to the participants on the screen with attribute questions (e.g. Experience) above and below it. The question at the top was about the person on the left side of the picture (e.g. Please rate the Experience (ability to feel) of the

person on the left), and the one at the bottom was referred to the person on the right side of the screen (e.g. Please rate the Experience (ability to feel) of the person on the right). Participants moved a slider to a whole number on a scale ranging from 0 (the lowest level) to 10 (the highest level) to answer each question. The trial order was randomized, and the participants could complete the experiment at their own pace, but took no longer than 5 min.

### 2.1.5. Data analysis

#### 2.1.5.1. Main analysis

Since this study is a replication of the Medusa effect, we analysed the data in the same way as Will *et al.* [2] did. We compared whether there were significant differences between participants' perceptions of L1 and L2 on the three dimensions of Realness, Agency and Experience based on paired *t*-tests. Confirmation of our hypothesis was based on the following criteria.

For H1, we predicted that participants will rate L1 (picture of a person) as having higher levels of Realness, Agency and Experience than L2 (picture of a picture of a person). Significantly higher scores ($\alpha = 0.05$) for L1 than L2 would indicate acceptance of H1 as well as successful replication of Will *et al.*'s [2] Experiment 2.

#### 2.1.5.2. Equivalence test

If Study 1 did not replicate Will *et al.*'s [2] Experiment 2 successfully, we would then conduct equivalence tests to examine whether the non-significant results provide evidence for the effect's absence or negligible size [26].

The smallest effect size of interest (SESOI) for our equivalence test was determined according to the small telescopes argument [27] as the effect size the original design had 33% power to detect. For Realness, based on a power analysis and considering the 106 participants of the original experiment, a one-side paired *t*-test with an alpha of 0.05 would have had 33% power to detect an effect of $d = 0.1178$. This would be taken as our SESOI for Realness. Similarly, the SESOI would be $d = 0.1167$ for Agency and $d = 0.1195$ for Experience.

# 3. Study 2

Study 2 employed a mind perception task and a dictator game task to examine whether pictorial abstraction levels affect conduct in social interactions, which was a replication of Will *et al.*'s [2] Experiment 5, in Japan.

## 3.1. Method

### 3.1.1. Independent variable

#### 3.1.1.1. Different levels of abstraction of people's photos

Our study had two abstraction levels: L1 and L2. Similar to Will *et al.*'s [2] study, pictorial abstraction was a within-subjects factor.

#### 3.1.1.2. Levels of abstraction of recipients in the dictator game

There were two levels of abstraction for the receiver in the dictator game, classified as L1 or L2. This was a between-subjects factor.

### 3.1.2. Dependent variable

#### 3.1.2.1. Mind perception

In Study 2, mind perception was measured through a 2AFC task that indicated which of the two individuals (L1 or L2) had higher attributes of Realness, Agency and Experience.

### 3.1.2.2. Amount of allocated money in the dictator game

In the dictator game, participants decided the amount of money they would like to allocate from 0 to 1000 Japanese yen to L1 person or L2 person.

### 3.1.3. Participants

#### 3.1.3.1. Sample size and power analysis

In Study 2, we obtained our sample size based on the given effect size Cohen's $d$ ($d = 0.36$) of the Wilcoxon-Mann–Whitney test in the original study, similar to Study 1. In line with the same principle to avoid possible existence bias in the replication study, we also used a small effect size $d$ value and conducted a non-parametric independent sample power analysis of one-tailed, normal parent distribution, assuming an effect size Cohen's $d = 0.2$, significance level $\alpha = 0.05$, power level $1 - \beta = 0.80$, to compute the sample size. In addition, considering the necessity of counterbalance, we decided to equalize the participants of the two groups by setting the allocation ratio N2/N1 to 1. The results revealed that 325 participants per group were required to obtain a power of 0.80. Essentially, there was no case of data not being collected by participant withdrawal since the agreement of participants granted for only one assessment-allocation task in Study 2. Nevertheless, 0.5% of the data loss occurred due to unknown failure in the original study, and approximately 1% of the data were excluded because of failed attention checks. Thus, we added an additional 1.5%, which was five people to each group in Study 2, and at least 660 participants (330 per group) would be recruited in total.

#### 3.1.3.2. Recruitment and screening

Study 2 used the same recruitment and screening methods as Study 1, except that participants who had completed Experiment 1 were excluded. The minimum sample size of Study 2 was 660 participants. Considering that it was difficult for us to limit the number of participants to exactly 660 due to the characteristics of the participatory online recruitment system, we recruited at least 660 participants and used their data for the analysis based on the timestamp.

#### 3.1.3.3. Stimuli and design

The original stimuli (pictures) had already been provided to us via email by the original authors. We employed the same stimuli as in the initial study. Four photographs of two experimental model volunteers (Person A and Person B) were presented as stimuli. These photographs consisted of two versions (original displays and horizontally inverted mirror displays) of two original photographs, balancing the levels of pictorial abstraction (L1, L2) and spatial location (left, right). For the original photographs, each captured the entire face of one model together with the life-sized photo of another model that she was holding. For example, the photograph depicted Person A (L1) holding a life-sized photo of Person B (L2), or Person B (L1) holding a life-sized photo of Person A (L2). Each of the four pictures had been cropped to 1800 pixels high by 2400 pixels wide, to be displayed on the screen during the experiment.

Following the original design of Will et al. [2], participants were randomly assigned to four versions of display (photograph) and completed the mind perception task, which was to decide which of two people (person A or person B) seemed to have higher Realness, Agency and Experience. Afterwards, the dictator game was conducted based on the same display, in which participants were randomly assigned the L1 person or the L2 person as the recipient, to examine the connection between pictorial abstraction and prosocial behaviour. Finally, we conducted an attention check to ensure that the data were valid.

### 3.1.4. Procedure

Study 2 strictly followed the same procedures used in Experiment 5 by Will et al. [2]. Participants were recruited in the same criteria, and the experimental statement administered to participants was the same as in Study 1.

First, in the mind perception task, participants were randomly shown one of four photographs, which depicted Person A and Person B. Participants completed a 2AFC task by answering three questions: which person seems more real? Which person seems to have more Agency (ability to do)? Which person seems to have more Experience (ability to feel)?

Afterwards, participants proceeded to a one-shot dictator game using the same display. They were randomly assigned to groups with L1 or L2 as the designated recipient, indicated by an onscreen

arrow and text instructions. The task was to share 1000 Japanese yen endowment with a specified onscreen recipient. A slider in yen (0–1000) was displayed at the bottom of the screen, and participants manipulated the slider to decide the amount of allocation to the recipient. After the allocation was made, the participants completed the final step as the attention check by selecting option four from a list of five options. Participants who failed the attention check were excluded.

### 3.1.5. Data analysis

Since Study 2 was a replication of Will *et al.*'s [2] Experiment 5, we analysed the data of the mind perception task and the dictator game in the same manner as Will *et al.* [2] did in their Experiment 5. For the mind perception task, we conducted a binomial test to compare the proportion of participants choosing L1 and L2. Subsequently, a Mann–Whitney test was conducted to compare money allocations between L1 recipients and L2 recipients in all four counterbalanced versions of the stimuli. Finally, we conducted a Mann–Whitney test to compare whether participants who perceived L1 as higher than L2 on all three dimensions in the mind perception task made a strong distinction between L1 and L2 in the money allocation of the dictator game.

For H2-a, we predicted that the proportion of participants choosing L1 will be higher than that of participants choosing L2 for all three dimensions (Realness, Agency and Experience) in the mind perception task. A proportion that is significantly above ($\alpha = 0.05$) the chance level of 50% will indicate acceptance of H2-a. Moreover, for H2-b, significantly more money ($\alpha = 0.05$) allocated to L1 recipients than L2 recipients will indicate acceptance of H2-b. As for H2-c, participants who perceived L1 as higher than L2 on all three dimensions and differentiated significantly between L1 and L2 ($\alpha = 0.05$) in their dictator game allocations will indicate acceptance of it. Acceptance for all three hypotheses of Study 2 will indicate the success of the replication of Will *et al.*'s [2] dictator game task in Experiment 5.

# 4. Study 3 (conditional study)

Study 3 is a conditional study comprising two parts, Studies 3-a and 3-b, and will be conducted only if H1 and/or H2 was not supported in Study 1 and/or Study 2. There may be two possible reasons for this failure. First, there is a possibility that the Medusa effect does not exist or only exists under very limited conditions; for example, the results may vary among different participants on account of their preferences for facial appearance. Secondly, the results may have been caused by limitations of the stimuli (e.g. quality and race). Considering the above reasons, we planned this conditional study (Study 3) by using newly designed stimuli with higher quality and the same ethnicity of participants to further replicate the original Experiments 2 and 5.

As for redesigning the stimuli, we would consider the factor of cross-cultural differences and further enhance the control of irrelevant variables. The original stimuli were pictures of Western people, which were different from our participants in Japan. Certain factors in the visual cognition of faces of different ethnicities (e.g. overall deterioration of identification/discrimination) could be the reason that the replication fails. Therefore, we would redesign the stimuli using Japanese faces. Moreover, instead of the pictures downloaded from the Internet in the original research, we would take pictures of volunteers in controlled conditions ourselves to exclude irrelevant variables, including, but not limited to, gender, expression, size and angle of the portraits.

## 4.1. Study 3-a

Study 3-a is a replication of Will *et al.*'s [2] Experiment 2 with newly created stimuli. If H1 is not supported in Study 1, we will conduct Study 3-a to further examine the existence of the Medusa effect. We hypothesize that participants would rate L1 as having higher levels of Realness, Agency and Experience than L2.

## 4.2. Method

### 4.2.1. Stimuli, design, participants and procedure

Similar to Study 1, the stimuli of Study 3-a will be 30 pictures; each picture depicts an L1 person and an L2 person, which present different degrees of abstraction in a single scene. For example, one person (L1) holds a portrait photograph (L2). We will design new stimuli by recruiting Japanese volunteers to capture

**Table 1.** Summary results for Study 1. Columns show the levels of abstraction rated by participants. Rows show the dimensions for rating. Cells show the mean rating scores that participants gave to L1 and L2 for each dimension.

| dimension | L1 | L2 |
|---|---|---|
| Realness | 8.64 | 6.07 |
| Agency | 6.89 | 4.88 |
| Experience | 6.33 | 5.55 |

their portraits of the entire face. Irrelevant variables, including but not limited to expression, size and angle, will be controlled while taking the pictures. Gender will be counterbalanced in redesigned stimuli.

The design, sample size, recruitment and screening, and procedure of Study 3-a will be the same as those of Study 1.

## 4.3. Study 3-b

Study 3-b is a replication of Will *et al.*'s [2] Experiment 5, except for the use of newly designed stimuli. If our Study 2 fails to replicate the original Experiment 5, Study 3-b will be conducted. The hypotheses of Study 3-b are the same as those of our Study 2.

## 4.4. Method

### 4.4.1. Stimuli, design, participants and procedure

Similar to Study 2, the stimuli in Study 3-b will be four photographs of two Japanese model volunteers (Person A and Person B) recruited in Japan. Each of these photographs captures the entire face of one model together with the life-sized photo of another model that she is holding. The photographs consist of two versions (original displays and horizontally inverted mirror displays) of the two original photographs, balancing the pictorial abstraction levels (L1, L2) and spatial location (left, right). For example, the photograph depicts Person A (L1) holding a life-sized photo of Person B (L2) or Person B (L1) holding a life-sized photo of Person A (L2). Irrelevant variables, including but not limited to expression, size and angle, will be controlled while taking the new pictures.

The design, sample size, recruitment and screening, and procedure of Study 3-b will be the same as that of Study 2.

# 5. Results

## 5.1. Study 1

All analyses conducted corresponded to Stage 1 peer-reviewed pre-registration (https://osf.io/qjwbx/). Based on the exclusion criteria and stopping rule, 158 participants were excluded from the analyses due to not completing the experiment, and 602 participants ($M = 50.17$, s.d. $= 11.50$, 150 female, 1 other/non-respondent) were analysed. The participants were randomly assigned to rate each of the two people in the picture based on Experience (198 participants), Agency (185 participants) or Realness (219 participants). Participant ratings are summarized in table 1. We used R version 4.1.3 [28] for all statistical analyses. Effect size was calculated by *effectsize* package [29] version 0.6.0.1.

Paired-sample Student's $t$-tests were conducted to compare the differences between participants' perceptions of L1 and L2 on the three dimensions of Realness ($t_{218} = 14.55$, $p < 0.001$, Cohen's $d = 0.98$), Agency ($t_{184} = 11.98$, $p < 0.001$, Cohen's $d = 0.88$) and Experience ($t_{197} = 6.97$, $p < 0.001$, Cohen's $d = 0.50$). On all three dimensions, participants rated L1 as having significantly higher levels of perception than L2. These results supported our H1, as well as indicating successful replication of Will *et al.*'s [2] Experiment 2.

## 5.2. Study 2

All analyses conducted corresponded to Stage 1 peer-reviewed pre-registration. Based on the recruitment criteria and stopping rule, a total of 794 participants in Japan completed the study online via the Yahoo!

**Table 2.** Summary results of the mind perception task in Study 2. Columns show the levels of abstraction compared by participants. Rows show the three dimensions of comparison. Cells show the percentage of participants arriving at each judgement.

| dimension | L1 | L2 |
|---|---|---|
| Realness | 89 | 11 |
| Agency | 81 | 19 |
| Experience | 81 | 19 |

Crowdsourcing Service. Nine participants were excluded because of failing to answer the attention check question, resulting in a final sample of 785 participants ($M = 52.12$, s.d. $= 11.44$, 174 female, 4 other/non-respondent). None of the participants participated in Study 1. Participants were randomly assigned to four counterbalanced versions of stimuli to complete the mind perception task. Then, participants were randomly assigned to the L1 dictator game ($n = 389$) or the L2 dictator game ($n = 396$).

### 5.2.1. Mind perception task

Participant choices are summarized in table 2. We conducted binomial tests to compare the proportion of participants choosing L1 and L2, which refer to our H2-a. The results confirmed that the proportions of participants choosing L1 were significantly higher than those of participants choosing L2 for all three dimensions (Realness: $p < 0.001$, 95% CI [0.87, 0.91]; Agency: $p < 0.001$, 95% CI [0.78, 0.84]; Experience: $p < 0.001$, 95% CI [0.78, 0.84]) in the mind perception task. The proportion of participants choosing L1 over L2 was significantly above the chance level of 50% for each dimension. Thus, our H2-a was supported.

### 5.2.2. Dictator game task

In the dictator game, participants were randomly assigned to L1 allocation condition or L2 allocation condition (0–1000 yen). Dictator game responses are summarized in figure 1.

The result of Mann–Whitney test showed that participants allocated significantly more money to L1 ($n = 389$, $M = 489.19$) than to L2 ($n = 396$, $M = 473.85$; Mann–Whitney $U = 84,716$, $z = 2.46$, $p = 0.01384$, Cohen's $d = 0.09$). Thus, our H2-b that L1 recipients will be allocated significantly more money than L2 recipients was supported. As for H2-c, participants who chose L1 rather than L2 on all three dimensions differentiated significantly between L1 and L2 in the dictator game allocations (Mann–Whitney $U = 55,890$, $p = 0.02849$, Cohen's $d = 0.068$). Therefore, our H2 were fully supported, indicating the success of the replication of Will *et al.*'s [2] Experiment 5.

## 6. General discussion

This present study successfully replicated the Medusa effect, which was the core finding of Will *et al.* [2], Experiments 2 and 5). In our registered replication, we again provided strong evidence that picture abstraction cost affects people's mind perception by recruiting participants of a different ethnicity (in Japan) from the original study (in the USA). Most crucially, L1 was rated as having a higher level of 'mind perception' than L2, which means that a 'picture of a person' was perceived as conveying more mind than a 'picture of a picture of a person' on all three dimensions (Realness, Agency and Experience). Following that, in our replication of the dictator game, we found again that representational abstraction of the photos also affects prosocial behaviour. As well as the choice of L1 being significantly higher than L2 in the results of the mind perception choice test, Japanese participants also allocated significantly more money to L1 than L2 recipients in the dictator game task, following the predictions. In other words, these results related behaviour to perception, which provided evidence to the effect that different pictorial abstraction levels influence behaviour in social interactions. Besides, as approximately 1400 participants were recruited in our Studies 1 and 2, these findings support the original claim's broad replicability and generalizability with a high power test.

For Study 1, the effect size of the present study was considerably larger than the original study on all three dimensions ($d = 0.98$ for the present study and $d = 0.83$ for the original study on Realness; $d = 0.88$

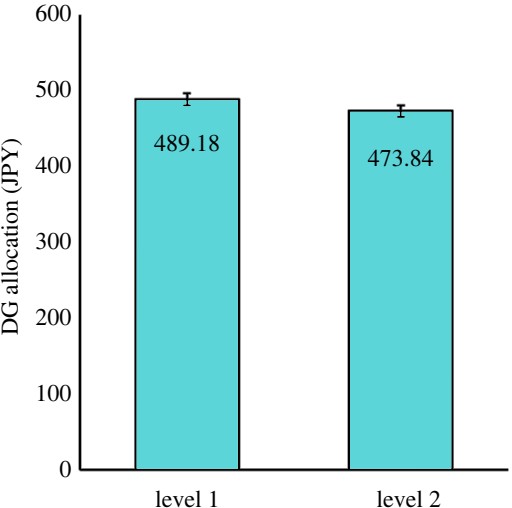

**Figure 1.** Results of the dictator game (DG) task in Study 2. Bars show mean money allocation to L1 and L2 recipients. Error bars show s.e.

for the present study and $d = 0.39$ for the original study on Agency; $d = 0.5$ for the present study and $d = 0.34$ for the original study on Experience). Moreover, since the sample size of our Study 1 was considerably larger than the original Experiment 2 (602 versus 320), the possible reason for the increase in which overinflation occurred due to a small sample size could be excluded. In other words, the observed increase of the effect size means that the participants of our study evaluated L1 as bringing stronger mind perception than L2, compared to participants of the original study.

As for Study 2, the previous findings were again validated in the mind perception choice test. That is, participants mostly perceived L1 as having higher levels of Realness, Agency and Experience than L2 in a statistically significant level. In the dictator game, there was a significantly higher allocation of money to L1 than L2 recipients, which was consistent with the results obtained in the original study. Nevertheless, the results clearly showed that the money allocation difference between the two groups in this study was smaller than the original study's allocation difference ($d = 0.09$ for the present study and $d = 0.36$ for the original study). Although we replicated the original Experiment 5 by conducting exactly the same procedure, there were some detailed differences between the two experiments, which could have led to the smaller disparity of the money allocation between two groups in our replication. Firstly, the sample size of our study was much larger than the original study (785 versus 202), and participants were recruited from Japan, not the USA. Regression to the mean due to the larger sample size than in the original study possibly could have reduced the resulting difference between the two groups. Besides race and sample size, gender proportion and average age of participants are also different between the present study and the original one. The proportion of female participants of our study was about 25%, which was less than 32% of Will et al.'s [2] Experiment 2. Doñate-Buendía et al. [30] found gender differences in giving decisions of dictator game, in which women were on average significantly more generous than men. Our lower proportion of female dictators may have led to the less allocation difference between L1 and L2 recipients, which was presented in the result as a smaller effect size. Regarding age, the average age of participants in our Study 1 was 50.17, due to the users' characteristics of the recruitment platform (Yahoo! Crowdsourcing Service). Although the age information of the participants is not available in the original study and cannot be accurately discussed, it is possible that this may have affected the results to some degree. Furthermore, since our Studies 1 and 2 successfully replicated Will et al.'s [2] Experiments 2 and 5, we did not perform Study 3, which was conditionally designed, following the criteria in the Stage 1 peer-reviewed pre-registration.

Constraints on the generalizability of the present study are as follows. The present study is a direct replication of the original research; we used exactly the same stimuli (out-group faces to Japanese participants) as in the original research. Our study confirmed the existence of the Medusa effect among participants from different cultures. However, further research may be helpful to better understand the progress of cross-culture effect on mind perception. Besides participants of different nationalities, investigations of stimulus culture are also essential for cross-culture effect clarification. Elfenbein et al. [31] highlighted the factorial experimental design, which is a balanced $n \times n$ across

**Table 3.** Study design and outcomes.

| question | hypothesis | sampling plan | analysis plan | rationale for deciding the sensitivity of the test for confirming or disconfirming the hypothesis | interpretation, given the different outcomes | outcome |
|---|---|---|---|---|---|---|
| Q1: does the level of pictorial abstraction affect mind perception? | H1: participants would rate L1 (person in picture) as having higher levels of Realness, Agency and Experience than L2 (person in picture of a picture) | 564 participants will be recruited in Study 1. The number of participants is based on a power analysis | the same as the original study; for H1, Study 1 uses three paired $t$-tests independently for each rating group to compare L1 and L2 | significantly higher scores in Realness, Agency and Experience of L1 as compared with those of L2 ($\alpha = 0.05$) would indicate the acceptance of H1 and the successful replication of Will et al.'s [2] Experiment 2 | If H1 is not supported, the replication of the Medusa effect fails. The results may be caused by the limitations of the stimuli (e.g. quality and race). We will redesign the stimuli with higher quality and the same ethnicity as the participants, and further try additional replications (Study 3-a) | H1 was supported (participants rated L1 as having significantly higher levels of Realness, Agency and Experience than L2) |
| Q2-1: does the level of pictorial abstraction affect mind perception (with stimuli that have better control of irrelevant variables)? | H2-a: participants would perceive L1 to be higher than L2 in all three dimensions (Realness, Agency and Experience) | 660 participants will be recruited in Study 2. The number of participants is based on a power analysis | same as the original study; for H2-a, we will use a binomial test to compare the proportion of participants choosing L1 and L2 | significantly higher ($\alpha = 0.05$) proportion of participants choosing L1 over L2 than the chance level of 50% will indicate acceptance of H2-a | If H2 is not supported, it would suggest that the Medusa effect is not related to prosocial behaviours. We will redesign the stimuli with the same ethnicity as the participants, and further try additional replications (Study 3-b) | H2-a was supported (participants perceived L1 to be significantly higher than L2 in all three dimensions) |
| Q2-2: does the level of pictorial abstraction affect conduct in social interaction? | H2-b: in the dictator game task, participants would allocate more money to L1 recipients than L2 recipients | | same as the original study; for H2-b, we will conduct a Mann–Whitney test to analyse the data of the dictator game task | significantly more money ($\alpha = 0.05$) allocated to L1 recipients than L2 recipients would indicate the acceptance of H2-b | | H2-b was supported (participants allocated significantly more money to L1 recipients than L2 recipients in the dictator game) |

(*Continued.*)

**Table 3.** (*Continued.*)

| question | hypothesis | sampling plan | analysis plan | rationale for deciding the sensitivity of the test for confirming or disconfirming the hypothesis | interpretation, given the different outcomes | outcome |
|---|---|---|---|---|---|---|
| Q2-3: can the effects in the dictator game be attributed to effects in the mind perception task? | H2-c: mind perception distinction would affect conduct in the dictator game | | same as the original study; for H2-c, we will conduct a Mann–Whitney test to compare participants' individual differences in perception and behaviour | participants who perceived L1 as higher than L2 on all three dimensions and differentiated significantly between L1 and L2 ($\alpha = 0.05$) in their dictator game allocations will indicate acceptance of H2-c | | H2-c was supported (participants who perceived L1 as higher than L2 on all three dimensions differentiated significantly between L1 and L2 in dictator game) |
| Q3-1: does the level of pictorial abstraction affect mind perception (with newly made stimuli)? | H3-1: participants would rate L1 as having higher levels of Realness, Agency and Experience than L2 | same as our Study 1. | same as our Study 1; for H3-1, Study 3-a uses tree paired *t*-tests independently for each rating group to compare L1 and L2 | significantly higher scores in Realness, Agency and Experience of L2 ($\alpha = 0.05$) would indicate the acceptance of H3-1 and the successful replication of Will et al.'s [2] Experiment 2 | If H3-1 is supported, it would suggest that the Medusa effect exists, and the race of stimuli may be the reason for the unsupported H1. If H3-1 is not supported, it would suggest that there is a possibility that the Medusa effect does not exist, or only exists under very limited conditions | since our Studies 1 and 2 successfully replicated Will et al.'s [2] Experiments 2 and 5, we did not perform Study 3, which was conditionally designed, following the criteria in the Stage 1 peer-reviewed pre-registration |
| Q3-2a: does the level of pictorial abstraction affect mind perception (with newly made stimuli)? | H3-2a: participants would perceive L1 to be higher than L2 in all three dimensions (Realness, Agency and Experience) | same as our Study 2 | same as our Study 2; for H3-2a, we will use a binomial test | significantly higher ($\alpha = 0.05$) proportion of participants choosing L1 over L2 than the chance level of 50% will indicate acceptance of H3-2a | If H3-2 is supported, it would suggest that the Medusa effect affects prosocial behaviours, the race of stimuli may be the reason for the unsupported H2. If H3-2 is not supported, it would suggest that the Medusa effect is not related to prosocial behaviours | |

*(Continued.)*

**Table 3.** (Continued.)

| question | hypothesis | sampling plan | analysis plan | rationale for deciding the sensitivity of the test for confirming or disconfirming the hypothesis | interpretation, given the different outcomes | outcome |
|---|---|---|---|---|---|---|
| Q3-2b: does the level of pictorial abstraction affect conduct in social interaction (with newly made stimuli)? | H3-2b: in the dictator game task, participants would allocate more money to L1 recipients than L2 recipients. | | same as our Study 2; for H3-2b, we will use a Mann–Whitney test | significantly more money allocated to L1 recipients than L2 recipients ($\alpha = 0.05$) would indicate the acceptance of H3-2b. | | |
| Q3-2c: can the effects in the dictator game be attributed to effects in the mind perception task (with newly made stimuli)? | H3-2c: mind perception distinction would affect conduct in the dictator game | | same as our Study 2; for H3-2c, we will use a Mann–Whitney test | participants who perceived L1 as higher than L2 on all three dimensions and differentiated significantly between L1 and L2 ($\alpha = 0.05$) in their dictator game allocations will indicate acceptance of H3-2c | | |

cultures design, to separate possible effects of stimulus culture from participants' nationalities. It has become a common and effective method to employ the balanced factorial design when discussing cross-cultural effect (e.g. [32–34]). For replication studies, for example, Yoshimura *et al*. [35] replicated a previous study investigating the influence of smiling on age estimation and examined the cross-cultural validity of this finding. They employed a factorial design with two participant groups (Japanese and Swedish) and two stimulus groups (Japanese and Swedish faces), yielding results indicating that stimulus culture and participants' nationalities had no discernible impact on age estimation In other words, they have successfully replicated the original study in a cross-cultural context, indicating the presence of a stable cross-cultural effect. Further registered replications involving simultaneous manipulation of stimulus culture and participants' nationalities are required to confirm the generalization of the Medusa effect. If further cross-culture replicated experiments succeed, it could indicate the reproducibility and generalizability of the Medusa effect, regardless of the nationalities of stimuli and participants.

Besides, the unit of currency in our study and the original study was different. We used the yen in the dictator game, whereas the dollar was used in the original study. With similar currency values, the numbers shown for the yen (0–1000) are much larger than those for the dollar (0–10) due to the exchange rate. Since both the units of two sliders were 1 in the money allocation task, the available choices of the amount in our experiment (1001) were much more than the original one (11). Further research is needed to discuss if the different granularity of scoring scales influence the results of money allocation.

Furthermore, although we did not conduct a replication of the original Experiments 3 and 4 corresponding to Stage 1 peer-reviewed pre-registration, it is necessary for future research to continually replicate these two Experiments to better understand the mechanism of the Medusa effect. Specifically, in the original Experiment 3, eye-tracking data show the progress of eye-movement during free viewing of stimuli. If differentiation between L1 and L2 occurs without an explicit compare instruction, it could reveal systematic differences in looking behaviour even in daily life. Moreover, replication in cross-cultural context could further provide eye-movement evidence of different races. It should be pointed out that own-race faces are recognized more accurately than other-race faces: the own-race bias (ORB) in face recognition (e.g. [36]). A number of researchers have discussed the causes of the ORB and found that people are more inclined to automatically process their own-race faces in depth, and are less motivated to do so when faced with other-race faces, leaving them at a superficial level of processing. Eye-movement data in cross-culture context would further examine the reproducibility, and explore if motivational differences in face processing due to racial differences affect the generalizability of the Medusa effect. As for the original Experiment 4, it will compare the transitions between different levels of pictorial representations among reality and pictures (real face versus picture of a face; picture of a face versus picture of a picture of a face), could further help with the application of the Medusa effect in daily life.

In conclusion, in the present study, we successfully replicated the results of Experiments 2 and 5 of Will *et al*.'s [2] as a registered report without any deviation from the protocol, and found the same results with the original stimuli among Japanese participants. Overall, our study confirmed the reproducibility of the Medusa effect and extended its generalizability in Japan. Further investigations would be required in future work to discuss characteristics of stimuli which could impact on abstraction costs of mind perception, such as content (e.g. facial expression and posture), format (e.g. separation or interaction) or presentation (e.g. in person, on screen and VR) (table 3).

**Ethics.** The current study has been committed by the ethics committee of Kyushu University (protocol number: 2022-015), and was conducted in accordance with the principles of the Declaration of Helsinki. The participants had the right to self-determine to cease the study at any time without any disadvantages. All participants were provided informed consent, and the study was started only with the granted approval. The personal information of all participants was strictly protected and will not be disclosed to third parties.

**Data accessibility.** Data and analysis code of this study are openly available on the Open Science Framework: https:// osf.io/qjwbx/[37].

**Declaration of AI use.** We have not used AI-assisted technologies in creating this article.

**Authors' contributions.** J.H.: conceptualization, formal analysis, investigation, methodology, writing—original draft, writing—review and editing; M.Z.: conceptualization, formal analysis, investigation, methodology, writing—original draft, writing—review and editing; J.L.: conceptualization, formal analysis, methodology, writing—original draft, writing—review and editing; Y.S.: conceptualization, formal analysis, investigation, methodology, writing—original draft, writing—review and editing; Y.Y.: conceptualization, formal analysis, funding acquisition, investigation, methodology, project administration, supervision, writing—original draft, writing—review and editing.

All authors gave final approval for publication and agreed to be held accountable for the work performed therein.

**Conflict of interest declaration.** Y.Y. is a recommender and managing board member at PCI Registered Reports.

**Funding.** This research is supported by JSPS KAKENHI: JP20H04581, JP21H03784 and JP22K18263.

**Acknowledgements.** We would like to thank Editage (www.editage.com) for the English language editing.

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
