## [Peer Review File · Royal Society Open Science]

Review History

RSOS-231802.R0 (Original submission)

Decision letter (RSOS-231802.R0)

Dear Dr Yamada

On behalf of the Editor, I am pleased to inform you that your Manuscript RSOS-231802 entitled "The Medusa effect: A registered replication report of Will, Merritt, Jenkins, and Kingstone (2021)" has been accepted in principle for publication in Royal Society Open Science. The reviewers' and editors' comments are included at the end of this email.

The admin team will now process the Stage 2 submission on your behalf.

You may now progress to Stage 2 and complete the study as approved. Before commencing data collection we ask that you:

- 1) Update the journal office as to the anticipated completion date of your study.
- 2) Register your approved protocol on the Open Science Framework (<https://osf.io/>) or other recognised repository, either publicly or privately under embargo until submission of the Stage 2 manuscript. Please note that a time-stamped, independent registration of the protocol is mandatory under journal policy, and manuscripts that do not conform to this requirement cannot

be considered at Stage 2. The protocol should be registered unchanged from its current approved state, with the time-stamp preceding implementation of the approved study design.

Following completion of your study, we invite you to resubmit your paper for peer review as a Stage 2 Registered Report. Please note that your manuscript can still be rejected for publication at Stage 2 if the Editors consider any of the following conditions to be met:

- The results were unable to test the authors' proposed hypotheses by failing to meet the approved outcome-neutral criteria.
- The authors altered the Introduction, rationale, or hypotheses, as approved in the Stage 1 submission.
- The authors failed to adhere closely to the registered experimental procedures. Please note that any deviations from the approved experimental procedures must be communicated to the editor immediately for approval, and prior to the completion of data collection. Failure to do so can result in revocation of in-principle acceptance and rejection at Stage 2 (see complete guidelines for further information).
- Any post-hoc (unregistered) analyses were either unjustified, insufficiently caveated, or overly dominant in shaping the authors' conclusions.
- The authors' conclusions were not justified given the data obtained.

We encourage you to read the complete guidelines for authors concerning Stage 2 submissions at <https://royalsocietypublishing.org/rsos/registered-reports#ReviewerGuideRegRep>. Please especially note the requirements for data sharing, reporting the URL of the independently registered protocol, and that withdrawing your manuscript will result in publication of a Withdrawn Registration.

Your feedback matters - please spend 5 minutes leaving anonymous feedback about your experience of Registered Reports at this journal, as an author or reviewer:
https://registeredreports.cardiff.ac.uk/feedback/feedback/decision_letter.php

This feedback is collected by the Registered Reports Community Feedback, website which is an independent service and research project, being undertaken by Cardiff University.

Once again, thank you for submitting your manuscript to Royal Society Open Science and we look forward to receiving your Stage 2 submission. If you have any questions at all, please do not hesitate to get in touch. We look forward to hearing from you shortly with the anticipated submission date for your stage two manuscript.

on behalf of Professor Chris Chambers (Registered Reports Editor, Royal Society Open Science)
openscience@royalsociety.org

Author's Response to Decision Letter for (RSOS-231802.R0)

See Appendix A.

RSOS-231802.R1 (Revision)

Decision letter (RSOS-231802.R1)

Dear Dr Yamada:

I am pleased to inform you that your Stage 2 Registered Report (PCI RR track) entitled "The Medusa effect: A registered replication report of Will, Merritt, Jenkins, and Kingstone (2021)" is now accepted for publication in Royal Society Open Science.

Please remember to make any data sets or code libraries 'live' prior to publication, and update any links as needed when you receive a proof to check - for instance, from a private 'for review' URL to a publicly accessible 'for publication' URL. It is also good practice to add data sets, code and other digital materials to your reference list.

Royal Society Open Science is a fully open access journal. A payment may be due before your article is published. Please note that, if the corresponding author of your paper is based at an institution covered by one of our Transformative Agreement deals, your fees may be covered by the deal – please check the list of eligible institutions at <https://royalsociety.org/journals/authors/read-and-publish/read-publish-agreements/>. The Royal Society has partnered with Copyright Clearance Center's (CCC's) RightsLink service to allow authors to pay article processing charges or page charges. After your manuscript has been accepted, the corresponding author will receive an email from CCC with the subject "Please submit your article processing/open access charge(s)/page charges" inviting you to pay your charges or request an invoice. The email from CCC will come from the email domain @copyright.com (if you have any queries regarding fees, please see <https://royalsocietypublishing.org/rsos/charges> or contact authorfees@royalsociety.org). If you request an invoice, it will be sent to you from CCC. It is important to be cautious about payment scams.

If you receive an email or text message requesting payment and have any concerns, we recommend contacting us through our website, rather than clicking on any links. The Royal Society will never ask you to make a direct payment.

Your feedback matters - please spend 5 minutes leaving anonymous feedback about your experience of Registered Reports at this journal, as an author or reviewer:
https://registeredreports.cardiff.ac.uk/feedback/feedback/decision_letter.php

This feedback is collected by the Registered Reports Community Feedback website, which is an independent service and research project, being undertaken by Cardiff University.

on behalf of Professor Chris Chambers (Subject Editor)

Follow Royal Society Publishing on Twitter: @RSocPublishing
Follow Royal Society Publishing on Facebook:
<https://www.facebook.com/RoyalSocietyPublishing/>
Read Royal Society Publishing's blog:
<https://royalsociety.org/blog/blogsearchpage/?category=Publishing>

Appendix A

This Registered Report was submitted to Royal Society Open Science following peer review and recommendation for Stage 2 acceptance at the Peer Community In (PCI) Registered Reports platform. Full details of the peer review and recommendation of the paper at PCI Registered Reports may be found at the links below.

After submission to the journal, the paper received no additional external peer review, but was accepted on the basis of the Editor's recommendation according to our PCI Registered Reports policy <https://royalsocietypublishing.org/rsos/registered-reports#PCIRR>.

This is a link to the Stage 2 recommendation and peer review history.

<https://rr.peercommunityin.org/PCIRegisteredReports/articles/rec?id=583>

This is a link to the Stage 1 recommendation and peer review history.

<https://rr.peercommunityin.org/PCIRegisteredReports/articles/rec?id=270>